# Research on a rapid identification method for counting universal grain crops

Jie Zhang[1], Shengping Liu[1,2], Wei Wu[1], Xiaochun Zhong (ID)[1,2☉]*, Tao Liu[3☉]

**1** Agricultural Information Institute, Chinese Academy of Agricultural Sciences, Beijing, P.R. China, **2** Key Laboratory of Agri-information Service Technology, Ministry of Agricultural, Beijing, P.R. China, **3** Jiangsu Key Laboratory of Crop Genetics and Physiology/Co-innovation Center for Modern Production Technology of Grain Crops, Yangzhou University, Yangzhou, P.R. China

☉ These authors contributed equally to this work.
* zhangjie_8902@163.com, zhongxiaochun@caas.cn

**Data Availability Statement:** All relevant data are within the paper and its Supporting information files.

## Abstract

Thousand-grain weight is a key indicator of crop yield and an important parameter for evaluating cultivation measures. Existing methods based on image analysis are convenient but lack a counting algorithm that is suitable for multiple types of grains. This research develops an application program based on an Android device to quickly calculate the number of grains. We explore the short axis measurement method of the grains with morphological thought, and determine the relationship between the general corrosion threshold and the short axis. To solve the problem of calculating the number of grains in the connected area, the study proposes a corrosion algorithm based on the short axis and an improved corner point method. After testing a variety of crop grains and equipment, it was found that the method has high universality, supports grain counting with white paper as the background, and has high accuracy and calculation efficiency. The average accuracy rate is 97.9%, and the average time is less than 0.7 seconds. In addition, the difference between the average accuracy for various mobile phones and multiple crops is small. This research proposes a grain counting algorithm with a wide range of applications to meet the requirements of non-glare use in the field. The algorithm provides a fast, accurate, low-cost tool for counting grains of wheat, corn, mung bean, soybean, peanut, rapeseed, etc., which is less constrained by space and power conditions. The algorithm is highly adaptable and can provide a reference for the study of grain counting.

## Introduction

Grain weight is important for improving crop yield [1]. Thousand-grain weight (TGW), which refers to the weight of a thousand seeds, is used to characterize the fullness of grains and is an important parameter for assessing cultivation measures [2]. In the breeding industry, TGW is a key indicator of the quality of a seed variety [3]. Therefore, TGW is widely employed in crop research [4–6]. Measuring TGW includes weighing and counting. The former is relatively simple and can be obtained directly through precision balance, while the latter is relatively cumbersome. The conventional methods of counting are presented as follows: (1)

**Funding:** National Natural Science Foundation of China 32172110,31701355 Mr. Tao Liu Fundamental Scientific Research Business Expenses of Central Public Welfare Research Institutes JBYW-AII-2020–29 Mr. Xiaochun Zhong Jiangsu Key Research and Development Program BE2020319-14 Mr. Tao Liu Special Fund for Independent Innovation of Agricultural Science and Technology in Jiangsu CX(21)3065, CX(21)3063 Mr. Tao Liu Innovation Project of the Chinese Academy of Agricultural Sciences CAAS-ASTIP-2016-AII Mr. Shengping Liu The funders were involved in the study and they played the following roles: (1)Xiaochun Zhong, conceptualization, funding acquisition, review. (2)Tao Liu, funding acquisition, methodology, resources, editing. (3) Shengping Liu, funding acquisition,supervision, review.

Manual counting is a traditional method that is time-consuming and laborious. Counting for a long time can be tiresome for people; therefore, they are more likely to make mistakes. (2) The infrared detection method can adapt to particles of different sizes and shapes [7, 8], its accuracy is high, but when the grains fall too fast, they are likely to be undercounted due to mutual obstruction. In addition, the cost is not low. (3) The method based on sound collision has a high accuracy [9] but is easily affected by the outside world, and the first installation is more complicated. (4) The method of image analysis is simple [10–13] the cost is also low, but the image segmentation algorithm poses difficulties. Such research includes two steps: (1) extracting the grain region from the image and (2) separating each grain region and then counting; this step is difficult. Common methods for solving these problems are described as follows: (1) Corrosion expansion method [14]. This method is simple to operate; the process begins by first separating the area of the adhesive with a corrosion operation and then performing the expansion to restore the grain size, enabling us to determine the connection boundary. However, different crop grains have different thresholds, and it is difficult to obtain a general segmentation value. (2) Watershed algorithm [15]. This widely employed method uses the local differences in the adhesion area to identify the split lines by simulating the flooding process of spring. This method is sensitive to noise and easily causes over-segmentation. (3) Active contour algorithm [16]. This algorithm uses the active contour model as the initial reference and then achieves segmentation of the target by gradually constructing the contour curve and approaching the edge of the object to be detected. The algorithm relies on the selection of the initial curve and needs to accurately identify the original properties of the grain. This method is only suitable for the adhesion of a small number of grains. (4) Feature point matching algorithm [17, 18]. Grain segmentation is realized by detecting feature points and calculating and matching descriptors, which is not suitable for grains with irregular edges. (5) Morphological algorithm [19, 20]. The principle is to extract the skeleton information of the grain image and then determine the posture and direction of the adhesion point through the operator. A smooth closing process is performed, first, according to the principle of direction, and second according to the principle of distance. Thus, it is possible to segment the image through the abovementioned process. This method is suitable for handling complex adhesion situations, but a large number of opening and closing operations result in a slow calculation speed.

Grain counting applications based on image segmentation mostly require the support of computers and cameras [21–23], and it is necessary to prepare the equipment in advance to perform a series of connection operations when using it. For example, Zhou Honglei used C++ and QT to develop a computer application. In addition, he designed a rice test platform to measure the number and area of grains, as well as other parameters, with 8 million pixel industrial cameras and metal indicator lights [21]. Yano developed SmartGrain software [11], which runs on a computer to measure the number and shape of high-throughput rice. Such methods are not very convenient. With the development of the mobile internet and smart terminals, people have become accustomed to using mobile phones to work and live. The Android platform provides a free development environment, which inspires a large number of practical agricultural applications (apps), such as agricultural pest identification [24], crop phenotype analysis [25], and plant species identification [26]. Some researchers have also conducted research based on app counting using mobile phones. For example, LiuTao proposed a method for calculating the number of grains based on image feature points [27], which could then accurately calculate the number of grains in the adhesion area. However, this method was only effective for rice and wheat grain recognition. Wu Wenhua designed the grainTKW system [28], which had a low cost and an accuracy rate of 97%, but it needed the support of powered equipment and brackets. Many variety tests were carried out outdoors and could not provide

powerful support for grain counting. To solve this problem, Evgenii Komyshev developed an outdoor counting app with white paper as the background [25], which could effectively identify the number of wheat grains. However, identifying 50 particles in 1 minute was too time-consuming for the requirement. This type of research predominantly focuses on one or two types of grains and lacks an algorithm suitable for a variety of crop grains. A fast, high-accuracy, low-cost, easy-to-operate counting method that supports outdoor environments and meets a variety of grain requirements is needed.

In summary, existing grain counting research is mostly carried out for a certain kind of grain. Some grain counting algorithms are not robust to various changes, and it is difficult to obtain a universal crop grain counting algorithm. Therefore, this research has developed a grain counting system that adapts to a wide range of grains. The GrainCounter app successfully enables users to quickly count the number of multiple types of outdoor grains through smartphones, thereby solving versatility and counting problems for the background of white paper. To eliminate the constraints of conditions, such as large space and power in laboratories, white paper, which is easy to obtain, is selected as the background. To solve the problem of identifying diverse grains of different colors, shapes, and sizes, a corrosion algorithm based on the short axis of the grain is proposed. To reduce GrainCounter's counting time, the research compresses pictures and executes the most time-consuming segmentation algorithm on the server.

## Materials and methods

### Grain varieties

To test the versatility of different grains, six grains of different colors, shapes, and sizes were selected: wheat, corn, mung bean, soybean, peanut, and rapeseed. Relevant descriptions are shown in Table 1:

GrainCounter was created for mobile devices. To test the accuracy of the algorithm for identifying grains under different device conditions, three Android operating system products with a high market share were selected: two mobile phones and a tablet. The parameters are detailed in Table 2. The test environments were indoors or outdoors under the shadow of a building or tree, on an A3 white paper, and in one test, the flash on the mobile phone used to take pictures was turned on. Because the tablet did not have a flash, the Huawei MLA-AL10 mobile phone was pasted on the back of the tablet, and we turned on the mobile flashlight to fulfill the light condition.

### Software development tools

The counting system was composed of three parts: the mobile terminal app, grain information management system, and grain recognition algorithm. (1) The app for the grain counting staff was developed by Java and Android Studio and communicated with the server through the

**Table 1. Description of grain types.**

| Grain | Color | Size (mm) | Shape | Variety |
|---|---|---|---|---|
| wheat | yellow | length: 7.2 width:3.4 | ellipse | Yangmai 23 |
| corn | yellow | length: 11.8 width:8.8 | Wedge | Zhengdan 958 |
| mung bean | green | length: 5.5 width:3.9 | ellipse | Zhonglv 1 |
| soybean | yellow | length: 11.1 width:6.9 | ellipse | Zhonghuang 13 |
| peanut | red | length: 13 width:8 | ellipse | Luhua 14 |
| rape | black | diameter:2.1 | round | Zhongshuang 11 |

**Table 2. Descriptions of test devices.**

| Manufactory | Product Type | Device Model | Version | Camera | Resolution |
|---|---|---|---|---|---|
| OPPO | Phone | R11s | 8.1.0 | CMOS 16 megapixels | 3456*4608 |
| HUAWEI | Phone | Honor Play COR-AL 10 | 9.1.0 | CMOS 16 megapixels | 3456*4608 |
| SAMSUNG | Pad | SM-T355C | 6.0.1 | CMOS 5 megapixels | 2592*1944 |

HTTP protocol. (2) The grain information management system served as the manager and was developed through Eclipse. The system belonged to the B/S structure and included the front end and back end. The former was based on the LayUI architecture and Jquery framework, and the latter employed the mainstream Spring MVC development model. Open-source MySQL was chosen as the database and was released through Tomcat. (3) The grain recognition algorithm was developed by OpenCV. After counting in this part, the number of grains, time, and picture after recognition are returned to the users. The system implementation flow is shown in Fig 1.

## Image capture

After clicking the upload photo button on GrainCounter, the app automatically called the camera to take pictures; a person could also select an image from the album. It was recommended to sprinkle grains on clean and flat white paper. The shooting direction of the camera was oriented perpendicular to the paper if possible. The use of the flash was recommended because it could reduce the influence of light. The shooting range enclosed all of the grains and contained a less irrelevant background (The sample image is shown in S1 Fig).

## Image process

The resolution of the original photo was so large that uncompressed operations would greatly increase the algorithm processing time, but excessive compression would reduce the algorithm accuracy. To balance the accuracy and response time, the following conventional resolutions were tested: 1080, 1440, 1920, 2500, and 3000. The numbers were the resolution numbers of the long side, and the short side was calculated in proportion to the original image. After the accuracy tests, we discovered that a resolution of 1920 worked best. Taking into account the different photographing habits of different people, horizontal and vertical photographs were captured. The study concluded that it did not affect the accuracy after the transposition test. Image compression and steering were processed using the open-source ImageMagick development kit.

## Grain separation

The study transformed the compressed original image into a grayscale image and then processed it into a binary image according to the threshold value to segment the research object and background area. In this study, we used the Gaussian filtering method to remove noise and utilized the OSTU algorithm to extract grain regions. The results are shown in Fig 2.

It was difficult to obtain a universal method to separate all kinds of seeds because the color, shape, and size of the seeds differed, and thus, achieving accurate separation by simple traditional methods was not easy. After analysis and exploration, a corrosion method based on short-axis grains, which could help conventional algorithms adapt to a narrow range of problems, was proposed in this research. This method is detailed as follows:

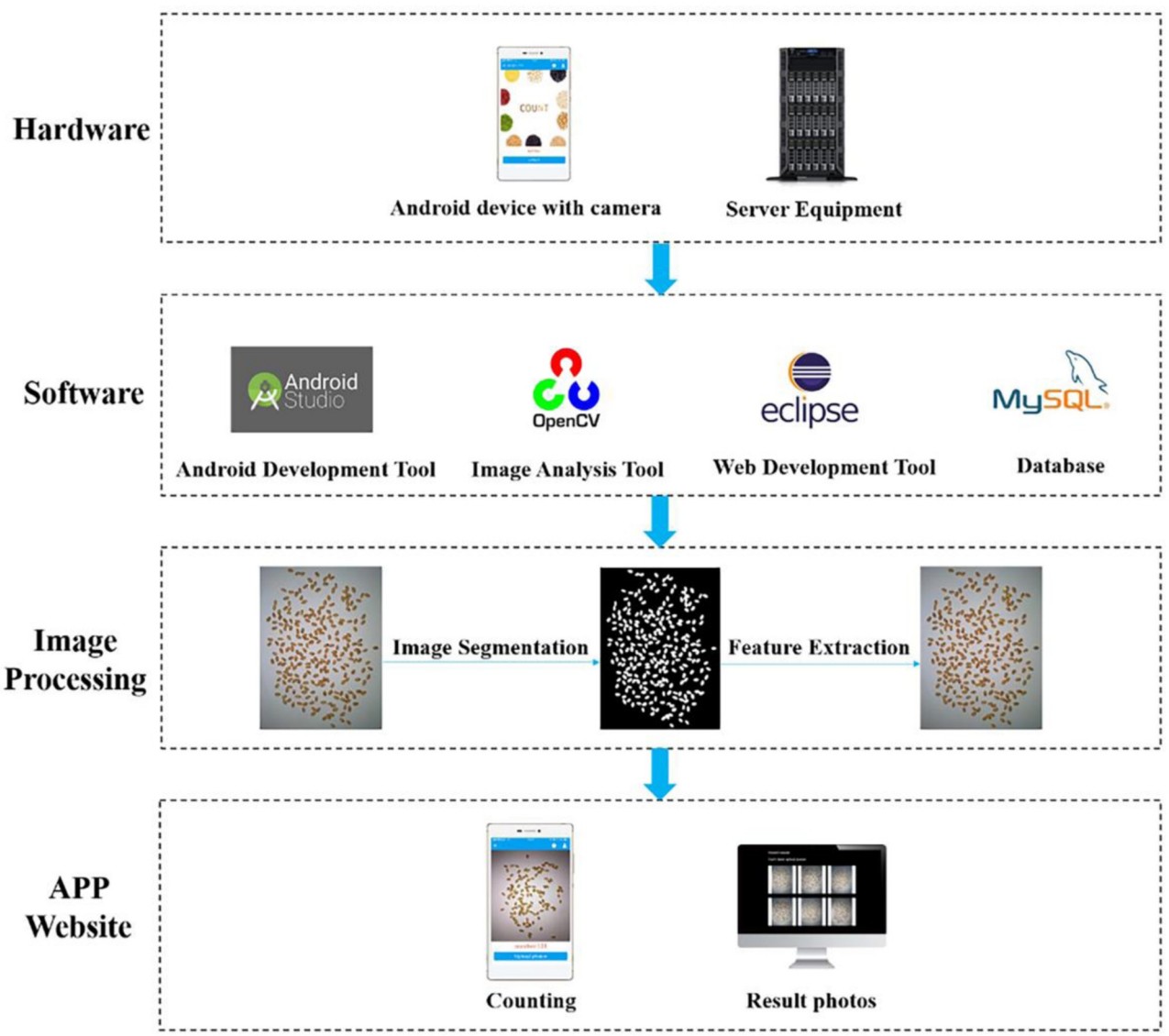

**Fig 1. System implementation flow.**

1. The grain skeleton was extracted through detailed operation, and then we calculated the grain skeleton length and distribution [29]. Fig 3(a) shows the results after treatment. The red line segment in Fig 3(b) denotes the grain skeleton, and the shorter line segment represents a single grain.

2. The short axis of the grain was calculated. First, we separately analyzed each point on a single grain skeleton, as points $P_1$, $P_2$, and $P_3$ are shown in Fig 4. We then calculated the vertical distance from the boundary to this point, as Lines $l_1$, $l_2$, and $l_3$ are shown in Fig 4. The maximum vertical distance of all points was the short axis of the grain.

3. Corrosion may break down the cohesive grains. If the corrosion was excessive, some grains could disappear. Therefore, the choice of parameters was very important. After conducting an analysis, we assumed that the degree of corrosion was related to the short axis of the grain. To meet the requirements of grains of different shapes and sizes, the corrosion

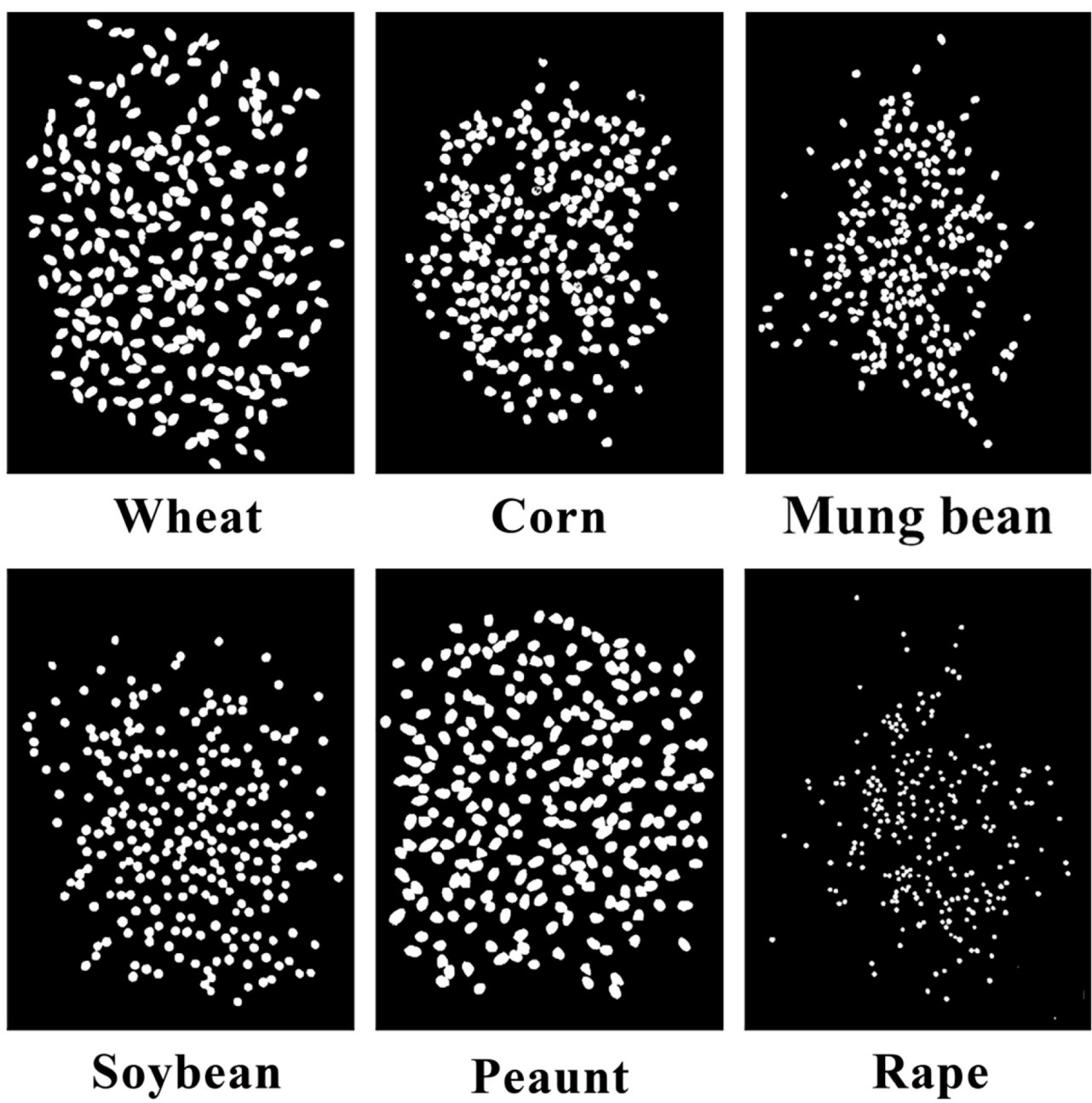

**Fig 2. Binary plots of 6 types of grains.**

parameter formula was set as:

$$e = w * X, \tag{1}$$

where w is the scale coefficient, and X is the short axis of the grain.

We took the values of 0.1, 0.2. . .. . .0.9 in this study and counted the proportion of unbonded grains and missing grains in the images. The analysis revealed that the best result was obtained when w was set to 0.4. Fig 5 shows the effects of 6 kinds of grains after corrosion of the 0.4*minor axis and 0.8*minor axis.

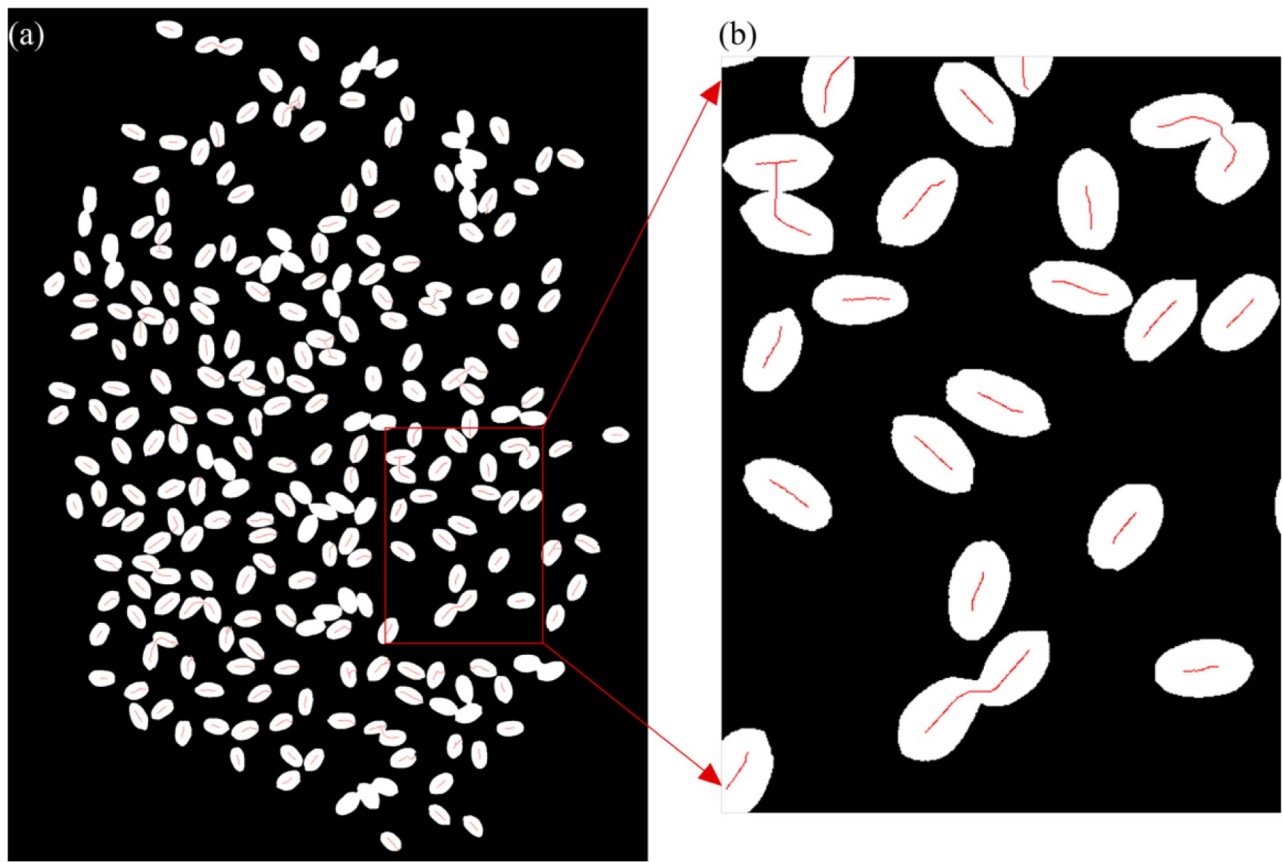

**Fig 3. Grain skeleton of wheat.** a: Grain skeleton graphics. b: Partially enlarged skeleton.

After segmentation with this method, the round grains were separated. Irregularly shaped grains could also decompose a large area of adhesion into a single grain or several small adhesion grains, as shown in Fig 5. Because a small number of adhesion areas still existed, we chose the corner point calculation method for analysis.

## Grain counting

**Improved corner calculation method.** By analyzing the grain position distribution, we found that it was mainly composed of three categories in Fig 6: (1) a single grain; (2) an area where the grain adhered; and (3) an area where the grain was adhered and had holes. It was difficult to achieve high accuracy in the case of grain adhesion for traditional methods, such as watersheds and active contours. To solve this problem, LiuTao proposed a corner point calculation method (Liu et al.,2017). In this method, the number of grains was calculated by the relationship between the feature points and the number of divided grains. The calculation formula is shown in (2):

$$Grain\ number \sum_{k=0}^{n} \frac{C_{peak}}{2} - R_{closed} + 1 \qquad (2)$$

where Cpeak is the local maximum proportion of the grain area in the template at the edge of the grain outline, and Rclosed is the closed area enclosed by the grain.

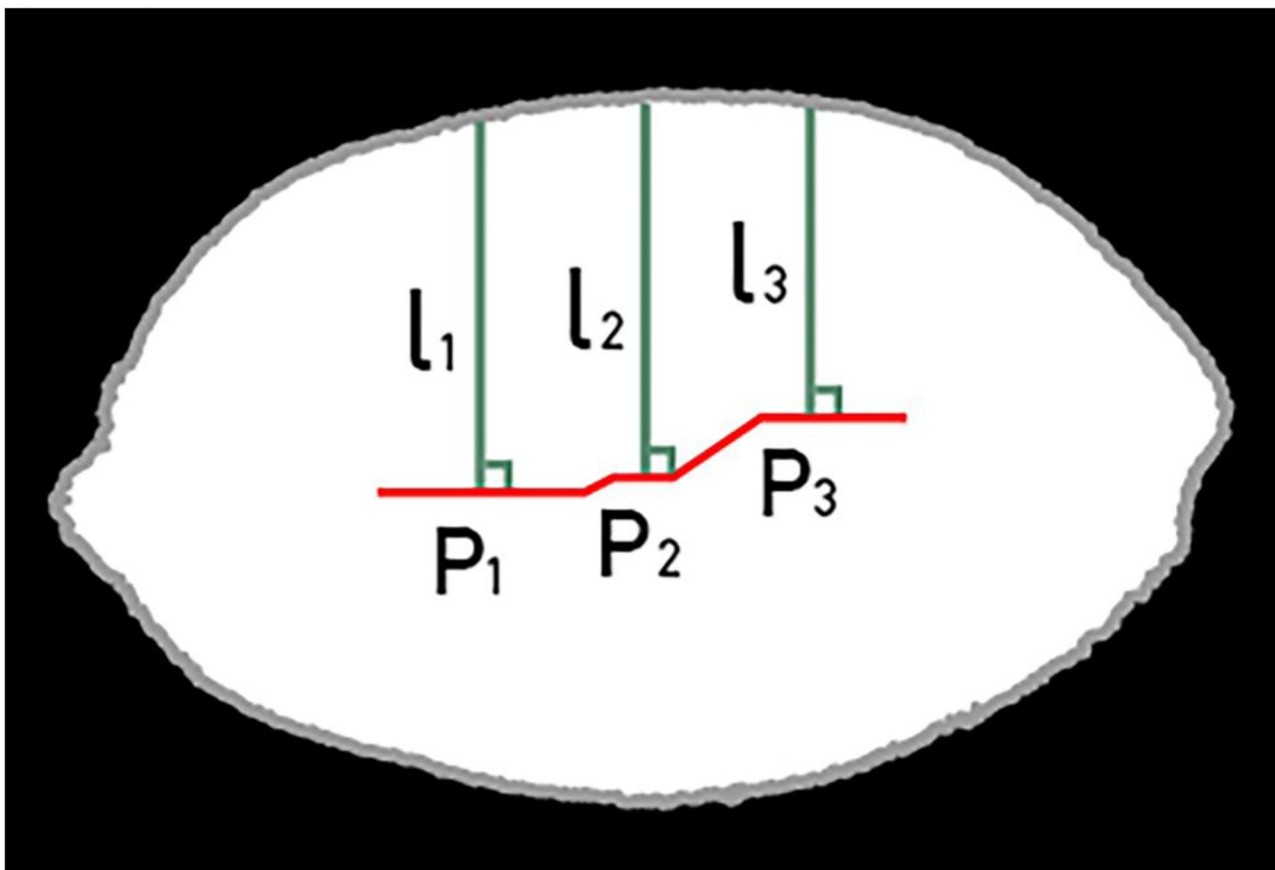

**Fig 4. The process of short-axis calculation.**

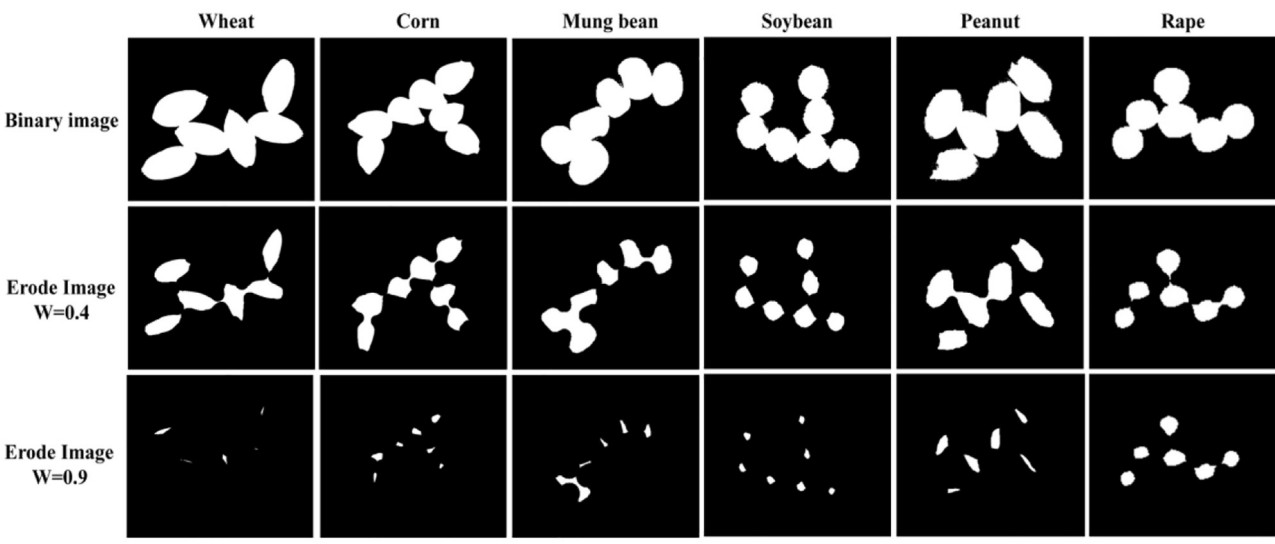

**Fig 5. Corrosion results of 6 kinds of grains based on different coefficients of short axis.** Note: The three rows are binary graphs; 0.4*minor axis of the corrosion factor, and 0.9*minor axis of the corrosion factor.

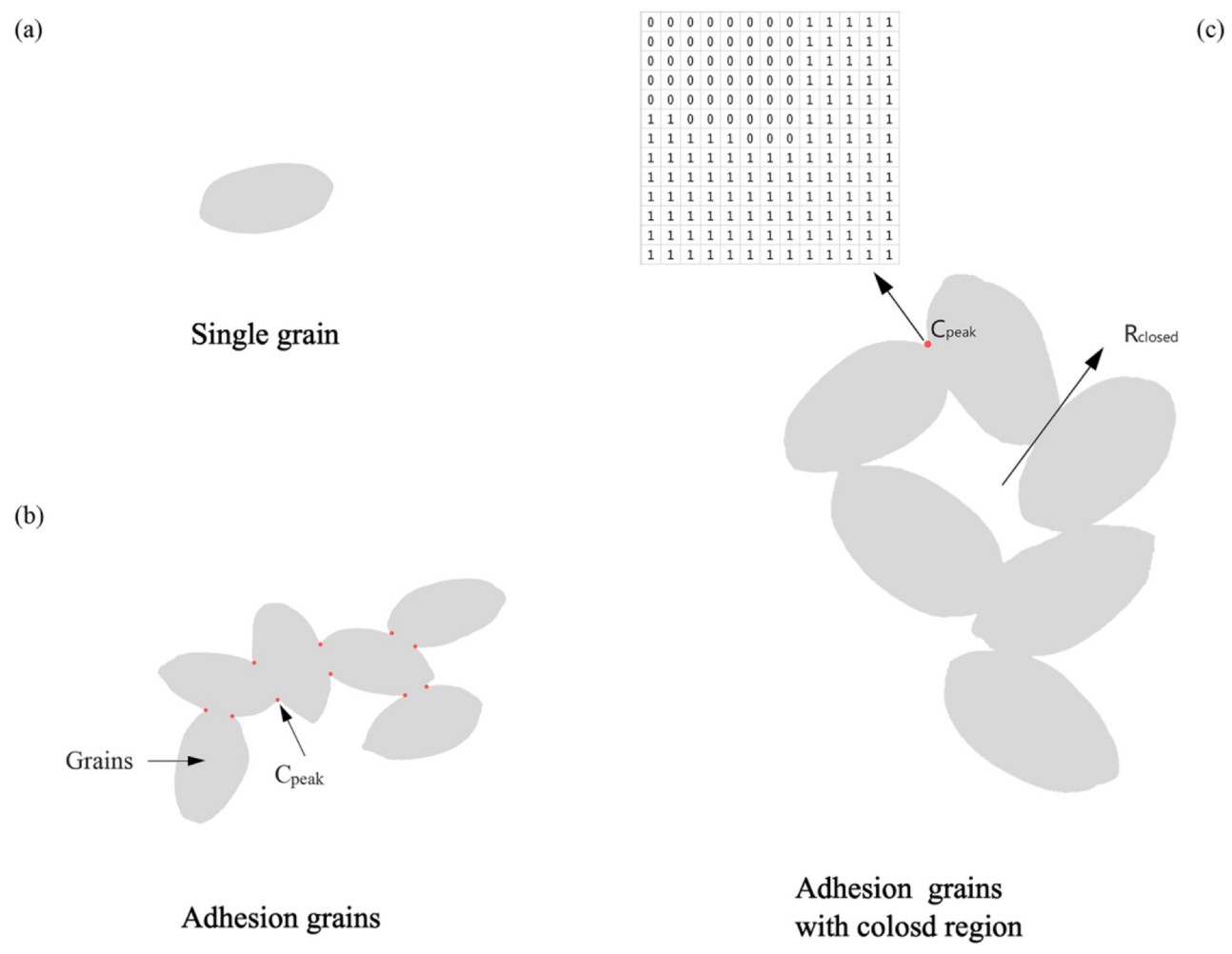

**Fig 6. Grain topology and corner points.**

After many experiments, we discovered that the given 5*5 neighborhood window in his research had some missed corners because the template was slightly small; if the template was too large, the algorithm processing time would be much longer, and it would be difficult to meet the application requirements. The results of the experiments in this study indicated that the neighborhood window was 13*13 better, as shown in Fig 6(c). The particle corner points were calculated using the parameter of 0.66*13*13 because this approach could provide more candidate corner points.

## Special perspective analysis

After many tests, the study found that the abovementioned corner point analysis method was prone to calculation errors when two grains were almost linearly connected. As shown in Fig 7, where there was no Cpeak point above due to overlap, and only the Cpeak point below was detected, and calculation result of the corner point algorithm was 1.5, which was actually 2 grains. Because oval-shaped grains, such as wheat, had irregular surfaces, the protruding parts were easily covered after overlapping, which might lead to missed detection of feature points.

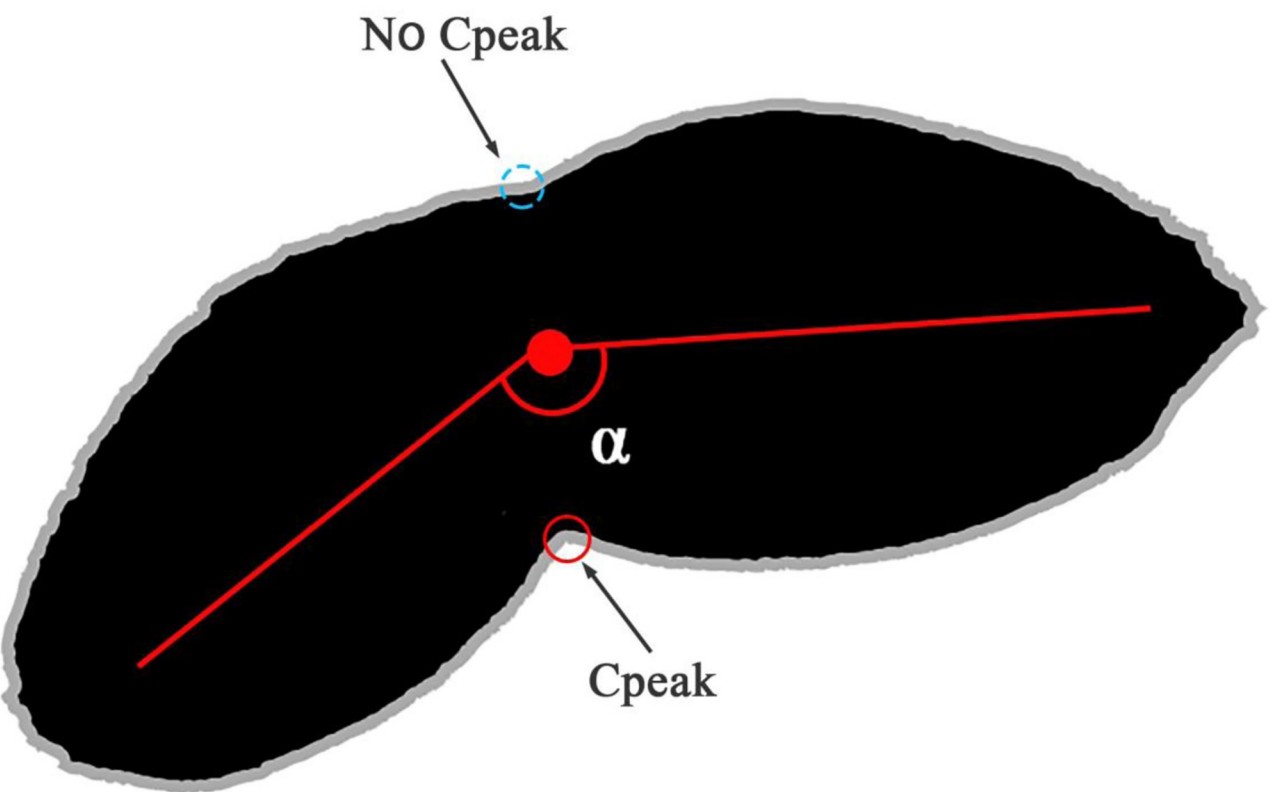

**Fig 7. Special topological relationship.**

Therefore, such results needed to be corrected. This research utilized morphological refinement operations to extract the skeleton and analyzed the length and angle of the skeleton line. According to the statistics, when the length of the skeleton line was greater than 1.5 times the average length and the included angle ranged from 160–200 degrees (clockwise), it was judged as a nonsingle seed. In this case, the average area method was used to calculate the number of grains in the adhesion area [30].

## Counting result display

The counting results are shown in Fig 8 and included in two parts. (1) Nonadhesive kernels. We chose a red dot to represent a grain; it was marked in the center of the grain, which was convenient for users to distinguish. (2) Adhesion of grains. It was represented by a red dot and a yellow number. The red dot, which was located in the center of the area, indicates that this part was an area composed of adhesive grains. The yellow number, such as numbers 3 and 2 in Fig 8(b), represents the number of grains in this area. The yellow number is marked at the upper right corner of the red dot, which is no longer involved in counting and just serves as an indication. The final number of the entire image = the number of red dots (excluding the red dots in the digital area) + the yellow number. For example, there were 7 grains in the picture (b). When the user saw this picture, it was easy to judge the accuracy of the algorithm. Observing a gap between the marked result and the actual situation, it was possible to correct it and record the adjusted result in the "corrected result" of the app.

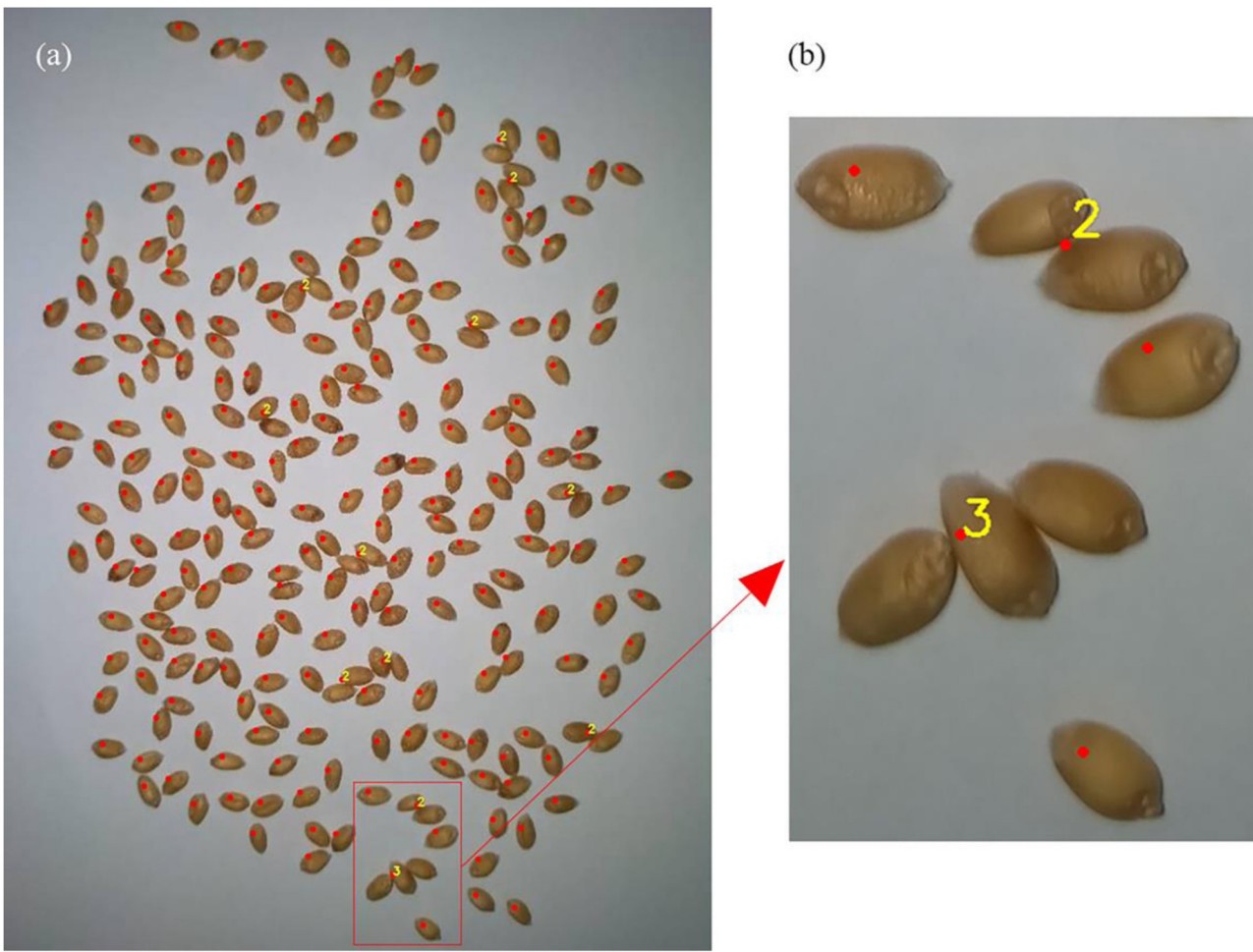

**Fig 8. Grain counting result.** Note. A: Counting result graph. B: Meaning of the red dots and numbers.

### Evaluation index

The correct ratio (CR), error ratio (ER), and volatility ($S^2$) were applied to evaluate the methods of this study. The corresponding formulas are shown in Eqs (3)–(6):

$$Correct\ ratio = \left(1 - \frac{|N_1 - N_2|}{N_1}\right) * 100\% \tag{3}$$

$$Error\ ratio = \left(\frac{|N_1 - N_2|}{N_1}\right) * 100\% \tag{4}$$

$$M = \frac{x_1 + x_2 + x_3 + \ldots + x_n}{n} \tag{5}$$

$$s^2 = \frac{(x_1 - M)^2 + (x_2 - M)^2 + (x_3 - M)^2 + \ldots + (x_n - M)^2}{n} \tag{6}$$

where N1 is the actual number of grains; N2 is the number of grains identified by the

algorithm; n is 8; xi is the accuracy of 8 kinds of grains; and M is the average identification accuracy rate of 8 groups, in which the grains are the same species, ranging from 50 to 400.

## Results and analysis

### Result evaluation

**Evaluation of the accuracy of different mobile phones and grains.** The number of counting grains in agriculture was concentrated in the range of 50–400 grains. Thus, a test program with these 2 numbers as the upper and lower limits and an arithmetic difference of 50 was designed. The test results are shown in Table 3 (The original statistical result is shown in S1 Table).

According to the statistical results, we could draw the following conclusions: (1) Comparing the average accuracy rate of each crop under three different equipment conditions, we found that corn grain had the lowest accuracy rate of 96.4% and that the grains of peanut and rapeseed had the highest accuracy rates of 98.9% and 98.8%, respectively. (2) The average accuracy rates of OPPO, Huawei, and Samsung tablets were 98.8%, 98.5%, and 96.3%, respectively. The identification accuracy of mobile phones was relatively high and relatively stable, with small differences in the results. The accuracy rate of Samsung's tablet was slightly lower than that of mobile phones and fluctuated greatly. The accuracy rate of identifying rapeseed grain was the

**Table 3. Test results of different equipment for 6 kinds of crops.**

| Grain Species | OPPO | | | | | HUAWEI | | | | | SAMSUNG | | | | |
|---|---|---|---|---|---|---|---|---|---|---|---|---|---|---|---|
| | GN | CR % | GN | CR % | AVG CR% | GN | CR % | GN | CR % | AVG CR% | GN | CR % | GN | CR % | AVG CR% |
| Wheat | 50 | 100.0 | 250 | 97.6 | 98.9 | 50 | 100.0 | 250 | 98.8 | 99.1 | 50 | 92.0 | 250 | 97.6 | 97.5 |
| | 100 | 99 | 300 | 99.7 | | 100 | 100.0 | 300 | 98.3 | | 100 | 94.0 | 300 | 98.3 | |
| | 150 | 99.3 | 350 | 97.4 | | 150 | 100.0 | 350 | 96.9 | | 150 | 92.7 | 350 | 90.9 | |
| | 200 | 98.5 | 400 | 100.0 | | 200 | 100.0 | 400 | 98.8 | | 200 | 95.0 | 400 | 95.8 | |
| Corn | 50 | 100.0 | 250 | 99.2 | 98.4 | 50 | 98.0 | 250 | 99.6 | 97.4 | 50 | 100.0 | 250 | 98.4 | 96.4 |
| | 100 | 100.0 | 300 | 99.0 | | 100 | 97.0 | 300 | 99.3 | | 100 | 98.0 | 300 | 96.3 | |
| | 150 | 99.3 | 350 | 98.3 | | 150 | 99.3 | 350 | 98.3 | | 150 | 95.3 | 350 | 92.3 | |
| | 200 | 98.0 | 400 | 93.3 | | 200 | 94.5 | 400 | 93.3 | | 200 | 83.5 | 400 | 83.5 | |
| Mung Bean | 50 | 100.0 | 250 | 98.8 | 98.8 | 50 | 98.0 | 250 | 99.2 | 98.4 | 50 | 98.0 | 250 | 99.2 | 97.8 |
| | 100 | 99.0 | 300 | 97.3 | | 100 | 100.0 | 300 | 96.3 | | 100 | 94.0 | 300 | 92.3 | |
| | 150 | 97.3 | 350 | 99.7 | | 150 | 97.3 | 350 | 98.9 | | 150 | 95.3 | 350 | 96.3 | |
| | 200 | 99.5 | 400 | 99.0 | | 200 | 99.0 | 400 | 98.5 | | 200 | 97.5 | 400 | 97.5 | |
| Soybean | 50 | 98.0 | 250 | 98.8 | 98.1 | 50 | 100.0 | 250 | 99.2 | 99.0 | 50 | 100.0 | 250 | 96.0 | 97.7 |
| | 100 | 99.0 | 300 | 97.7 | | 100 | 100.0 | 300 | 98.7 | | 100 | 98.0 | 300 | 93.0 | |
| | 150 | 100.0 | 350 | 97.1 | | 150 | 100.0 | 350 | 97.4 | | 150 | 99.3 | 350 | 93.1 | |
| | 200 | 99.0 | 400 | 95.3 | | 200 | 99.0 | 400 | 97.8 | | 200 | 95.5 | 400 | 93.0 | |
| peanut | 50 | 98.0 | 250 | 99.2 | 99.1 | 50 | 100.0 | 250 | 99.6 | 99.2 | 50 | 100.0 | 250 | 98.4 | 98.9 |
| | 100 | 100.0 | 300 | 98.7 | | 100 | 99.0 | 300 | 99.0 | | 100 | 100.0 | 300 | 98.3 | |
| | 150 | 100.0 | 350 | 97.4 | | 150 | 100.0 | 350 | 98.3 | | 150 | 98.7 | 350 | 95.4 | |
| | 200 | 100.0 | 400 | 99.5 | | 200 | 98.5 | 400 | 99.0 | | 200 | 99.5 | 400 | 96.5 | |
| Rape | 50 | 100.0 | 250 | 100.0 | 99.2 | 50 | 92.0 | 250 | 98.0 | 97.9 | 50 | 100.0 | 250 | 99.6 | 98.8 |
| | 100 | 100.0 | 300 | 98.3 | | 100 | 99.0 | 300 | 98.3 | | 100 | 100.0 | 300 | 99.7 | |
| | 150 | 98.7 | 350 | 97.4 | | 150 | 98.7 | 350 | 98.3 | | 150 | 100.0 | 350 | 97.4 | |
| | 200 | 99.0 | 400 | 100.0 | | 200 | 99.0 | 400 | 100.0 | | 200 | 99.0 | 400 | 99.5 | |

Note. GN means grain numbers, CR means correct ratio, and AVG CR means average correct ratio.

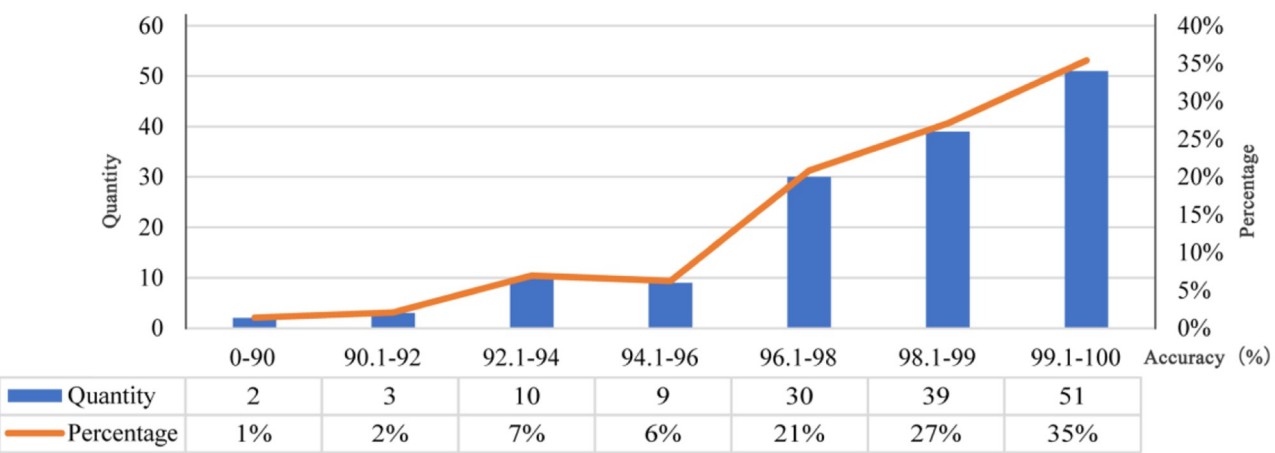

**Fig 9. Accuracy distribution of test grains.** Note. The horizontal axis is the accuracy rate of recognition by the algorithm; the left longitudinal axis is the quantity, and the right longitudinal axis is the percentage.

best, with an accuracy rate of 99.4%. Moreover, the accuracy rate of identifying corn grain was 93.4%, which was the lowest. (3) Overall effect: Using interregional statistical methods to analyze the distribution of the accuracy of different equipment and crop grains, the results are shown in Fig 9. The chart reflected that the 35% accuracy rate was concentrated in the range 99.1%-100%, followed by the range 98.1%-99% and the range 96.1%-98%, and accounted for 27% and 21%, respectively. Grains with an accuracy rate of more than 96.1% accounted for 83% of the total. There were 5 cases where the accuracy rate was less than 92%, accounting for 3%. (4) Special circumstances: There were two cases where the accuracy rate was lower than 90%, both of which were photos of 200 grains of wheat and 400 grains of wheat taken by Samsung tablets, and the counting accuracy rate was 83.5%.

## Identification efficiency evaluation

The time consumption required to recognize a picture based on a white paper background is shown in Fig 10. (1) The average running times of the OPPO, Huawei, and Samsung tablets were 0.647 s, 0.604 s, and 0.36 s, respectively. The time-consuming difference between the two mobile phones utilized in the test was small, and the time required for the tablet was the least. (2) The running time of the algorithm rarely exceeded 1 second. (3) The analysis time did not maintain a linear positive correlation with the number of grains in the picture. Various factors affected the running time, such as the tightness of the grains, size of the picture, and shape of the grains [25, 27, 28]. (4) Among the six test grains, the time to analyze corn was slightly longer and that of rapeseed grain was relatively short.

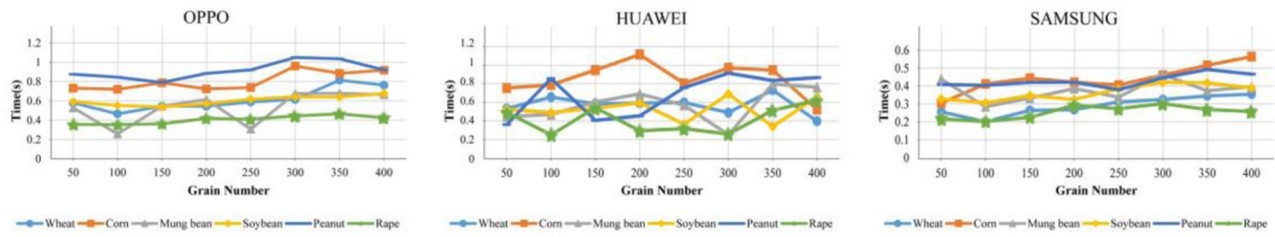

**Fig 10. Time-consuming algorithm.**

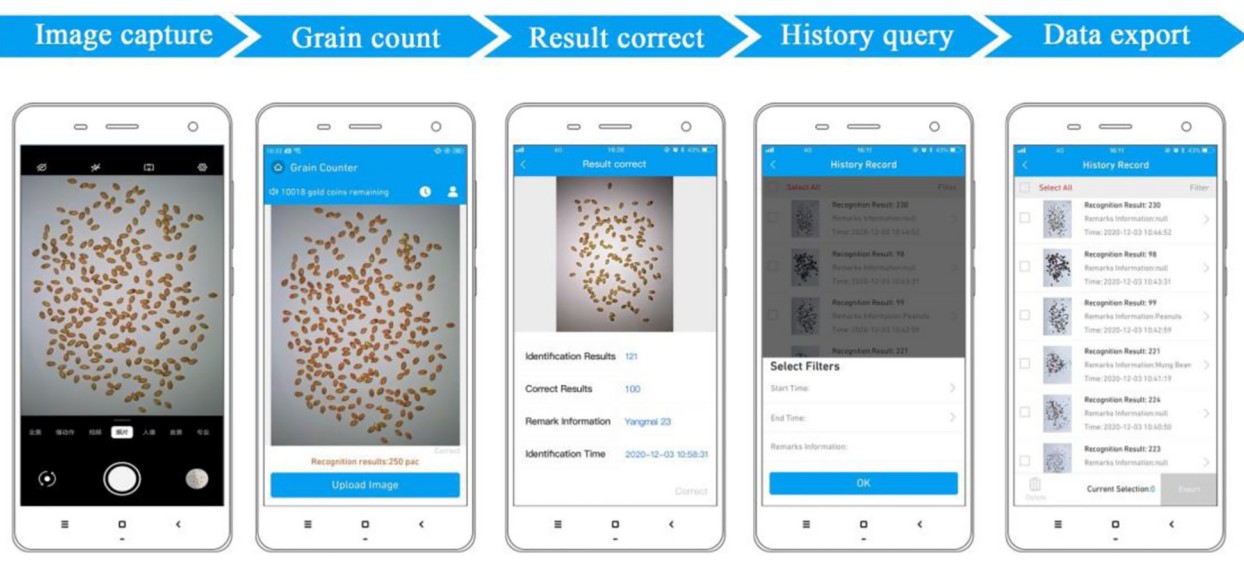

**Fig 11. Flow-chart diagram of the system.**

## Software main interfaces

The app was simple to operate, could meet the diverse needs of users, and had been commercialized. The main interfaces of the app are described as follows: (1) Through the "upload image" button, the users could directly take a picture or select the picture to be identified from the album, and the program automatically calculates the number. When a missed test or error was detected, the number of corrections could be entered with the "I want to correct" button in the lower right corner. (3) All historical results could be queried by clicking the clock icon. With the filter button, we could quickly search by time range and keywords of the remarks. (4) We checked the content to be exported and automatically sent the records to the specified mailbox in Excel format. (5) The application was very inexpensive; the single costs did not exceed 0.1 yuan, and the user could recharge the memory and issue an invoice. The flow-chart diagram of the system was shown in Fig 11.

## Discussion

### Comparison of the accuracy of different algorithms

It is common to use corrosion and watershed algorithms to calculate the number of grains. This study tests their accuracy on 6 kinds of selected grains, as shown in Table 4. The corrosion algorithm has relatively high accuracy on wheat, mung bean, and rapeseed grains, with an

**Table 4. Algorithm accuracy comparison.**

| Grain | Erode accuracy | Watershed accuracy | Algorithm in this paper |
|---|---|---|---|
| Wheat | 92.28% | 76.53% | 98.9% |
| Corn | 82.28% | 72.66% | 98.4% |
| Mung Bean | 91.75% | 89.60% | 98.8% |
| Soybean | 84.14% | 78.78% | 98.1% |
| Peanut | 81.60% | 84.73% | 99.1% |
| Rape | 93.15% | 97.16% | 99.2% |

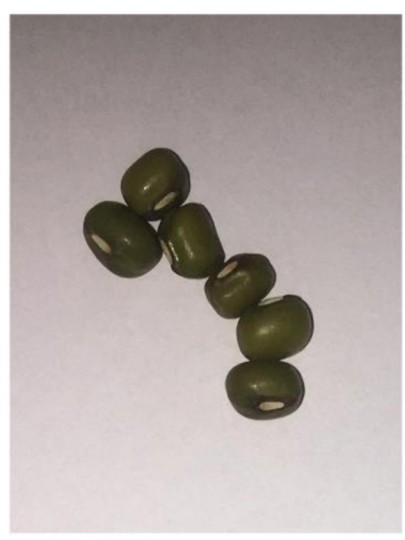
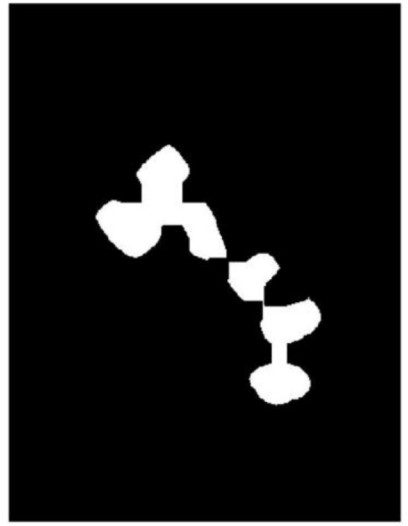
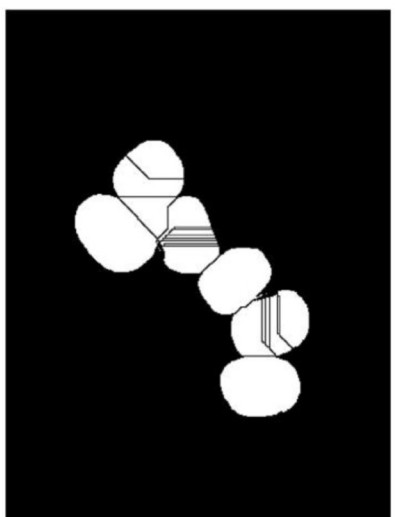

**Original Image**          **Erode Result**          **Watershed Algorithm**

**Fig 12. Results of corrosion and watershed analysis method of mung bean grains.**

average accuracy rate higher than 90%. The watershed algorithm is suitable for rapeseed grain, with an average accuracy rate of 97.16%. In other grains, the accuracy rate is relatively low.

Fig 12 shows the effects of these two methods for segmenting mung beans. Traditional corrosion algorithms are highly efficient, but it is difficult to select a separation threshold suitable for multiple crops. The watershed algorithm is easy to over segment. As shown in Fig 12, there are 6 grains of mung bean, but the calculated result of the watershed algorithm is 16 grains, producing a relatively large error rate.

The calculation of grains in the adhesion area is difficult in this research. The corner point algorithm has a satisfactory effect in identifying the grains that are adhered to each other and have smooth edges. However, for grains with irregular surfaces, it is easy for too many false corners to appear, which can cause a decrease inaccuracy. As shown in Fig 13(a), the blue points P2, P6, P7, P8, and P13 are the false corners of the corn kernels, and the red points are the real corners. The appearance of false corners would miscalculate the actual 7 kernels as 9.5 grains. In addition, when the grains in the adhesion area are oriented at a special angle, the true corners can be blocked, leading to a misjudgment. As shown in Fig 13(b), the protruding vertices of corn adhere to other grains, such as P1, P2, and P3. The original two corner points are mistaken for one corner point, which leads to calculation errors. Therefore, the accuracy of Liutao's algorithm [27] is reduced in these cases. As a result, we reduce the number of areas of adhered grains and then apply the corner point algorithm for grains that are very difficult to decompose. Fig 13(c) and 13(d) show the results of the algorithm processing in this research, which could provide a better representation of the impact of this type of combination.

### Efficiency comparison

Manually counting 400 grains, the traditional manual method takes approximately 140 s; the counting machine with mechanical rotation takes 24 s [31], and the wseen seed tester based on a personal computer (PC) and the camera takes 2.4 s. The smartphone app GrainTKW with the blue baseboard as the background takes 6.7 s-7 s [28]. The monochromatic board-based

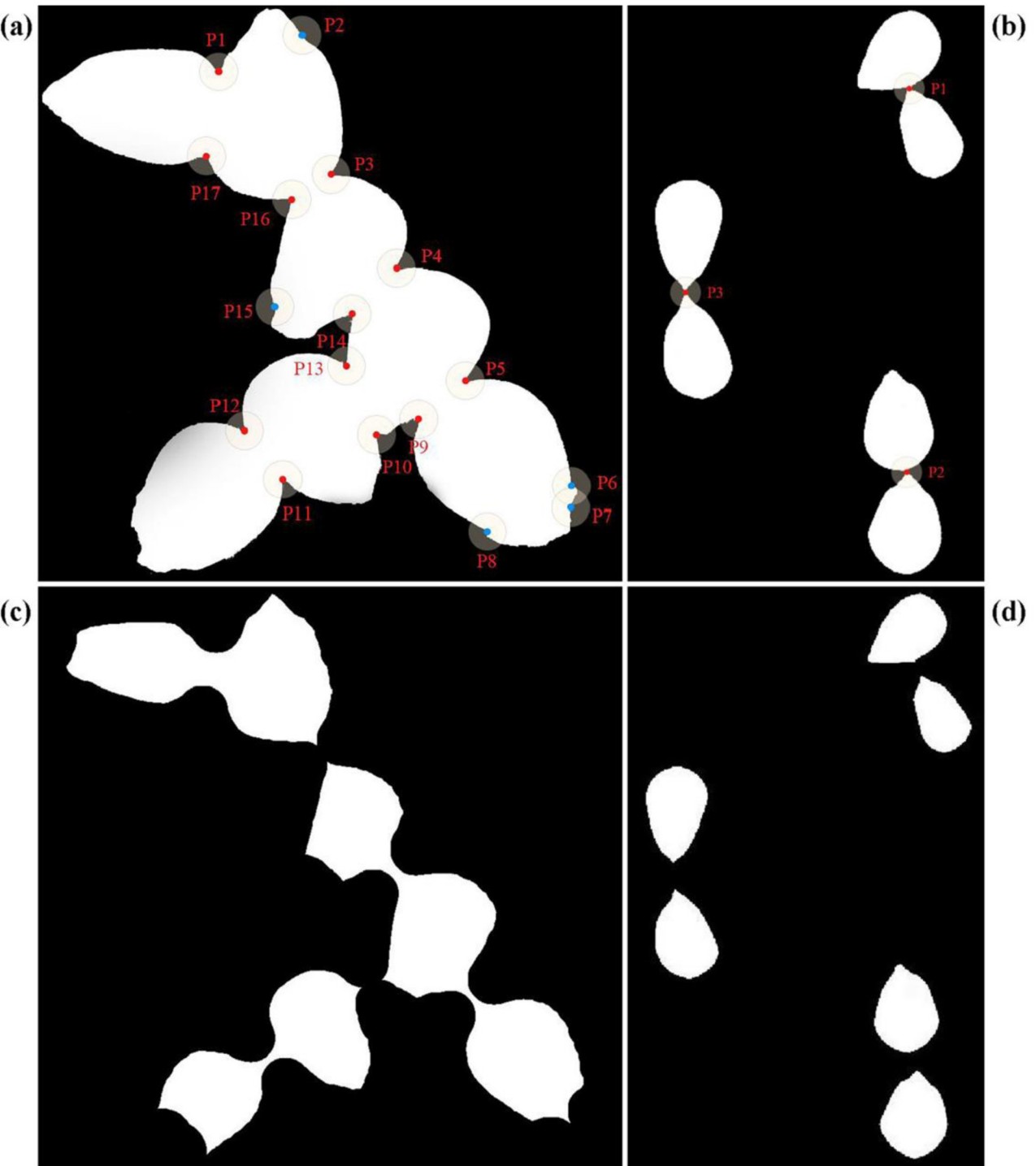

**Fig 13. Adhesion area analysis.** Note The distribution of grain corner points and the number of corner points under special position. C and D are the results of applying the algorithm of this paper to the two cases of A and B, respectively.

app developed by Liu Tao takes 1 s [27]. Dmitry Afonnikov developed the SeedCounter app based on Android phones and white paper, which takes 20–555 s to recognize 50 grains, and the algorithm takes 5–110 s [25]. In this study, the counting time of GrainCounter for 400 grains is approximately 4 s. Therefore, the algorithm takes less than 1 s. Approximately 3 s is needed to transfer pictures to the server and return the counting results. In terms of efficiency, the algorithm is relatively superior for the same type of products.

## Current kernel calculation tools

The current grain counting equipment is divided into mechanical equipment and electronic equipment. The former is mostly desktop equipment, such as Deren Electronics' grain counters, including hosts, splicers, and unloaders [31]. The total weight of the equipment is 41 kg; it is difficult to move and is only suitable for specific spaces. The latter is mostly based on image recognition technology, such as wseen's SeedTester [32]. This type of equipment requires a computer, imaging device, lighting box, and imaging board. In addition, the equipment is cumbersome to assemble, occupies a space of 40 cm×35 cm×40 cm, and is expensive. Small and medium enterprises and individuals cannot afford it. Mobile devices, such as mobile phones and tablet PCs, are highly popular, compact, and flexible. Although some researchers have developed apps based on mobile devices [27, 28], they require additional configuration and are generally only suitable for a small number of types of grains. The white paper is very easy to obtain, and this algorithm meets the technical requirements of a variety of grains. Therefore, the GrainCounter in this study, which is based on white paper and mobile devices, is more suitable for low-cost and wide-ranging counting requirements.

## Conclusion

This research introduces a grain counting system based on the background of a white paper. After testing six kinds of grains of different colors, shapes, and sizes, the grain counting algorithm in this study has a wide range of applications and can provide a fast, accurate, low-cost, easy-to-operate, and universal grain counting tool for breeders and precision cultivators. The tool in this research can calculate the number of grains under the condition of white paper outdoors, eliminate the constraints of indoor power supply, and meet the scenarios of low investment in rural breeding, low education level, and lack of manpower. After testing on mobile phones and a tablet PC, the average accuracy rate ranges from 96.4%-98.9%, and the average time to run the algorithm is less than 0.7 s.

## Supporting information

**S1 Fig. Sample image_of HUAWEI-Corn-WhitePaper-50.**
(JPG)

**S1 Table. The original statistical result of the sample grains.**
(XLSX)

## Author Contributions

**Conceptualization:** Xiaochun Zhong.

**Data curation:** Jie Zhang, Wei Wu.

**Formal analysis:** Jie Zhang, Wei Wu.

**Funding acquisition:** Shengping Liu, Xiaochun Zhong, Tao Liu.

**Investigation:** Jie Zhang.

**Methodology:** Jie Zhang, Tao Liu.

**Resources:** Tao Liu.

**Software:** Jie Zhang, Wei Wu.

**Supervision:** Jie Zhang, Shengping Liu.

**Validation:** Jie Zhang, Wei Wu.

**Visualization:** Jie Zhang.

**Writing – original draft:** Jie Zhang.

**Writing – review & editing:** Shengping Liu, Xiaochun Zhong, Tao Liu.

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
