## [Decision Letter · Decision Letter 0]

16 May 2022

PONE-D-21-36225Research on a rapid identification method for counting universal grain crops：a fast, accurate, low-cost, easy-to-operate, and universal grain counting toolPLOS ONE

Dear Dr. Xiaochun Zhong,

Thank you for submitting your manuscript to PLOS ONE. After careful consideration, we feel that it has merit but does not fully meet PLOS ONE’s publication criteria as it currently stands. Therefore, we invite you to submit a revised version of the manuscript that addresses the points raised during the review process.

           ACADEMIC EDITOR:The manuscript is dealing with very important subject that is extracting data using images. II have myself also read the manuscript and found this very useful for the scholars/students working in the area. The manuscript needs improvement by keeping in view comments of the reviewer. Some of my own comments are:1. Improve presentation of data in tables and figures2. The different steps involved may be presented in the form of flow chart3. Test files may be provided in the form of supplementary material4. Language and grammatical issues could be rectified. ==============================Please submit your revised manuscript by Jun 30 2022 11:59PM. If you will need more time than this to complete your revisions, please reply to this message or contact the journal office at plosone@plos.org. Please include the following items when submitting your revised manuscript:A rebuttal letter that responds to each point raised by the academic editor and reviewer(s). You should upload this letter as a separate file labeled 'Response to Reviewers'.A marked-up copy of your manuscript that highlights changes made to the original version. You should upload this as a separate file labeled 'Revised Manuscript with Track Changes'.An unmarked version of your revised paper without tracked changes. You should upload this as a separate file labeled 'Manuscript'.

We look forward to receiving your revised manuscript.

Kind regards,

Reyazul Rouf Mir, PhD

Academic Editor

PLOS ONE

**Journal requirements:**

“Xiaochun ZHONG, JBYW-AII-2020–29，Fundamental Scientific Research Business Expenses of Central Public Welfare Research Institutes，http://www.mof.gov.cn/index.htm， Conceptualization, Funding acquisition, Review

TaoLIU, 32172110,31701355, National Natural Science Foundation of China, https://www.nsfc.gov.cn/english/site_1/index.html,Funding acquisition, Methodology, Resources, Editing

TaoLIU, BE2020319-14, Jiangsu Key Research and Development Program, http://czt.jiangsu.gov.cn/, Funding acquisition, Methodology, Resources, Editing

TaoLIU, CX(21)3065，CX(21)3063, Special Fund for Independent Innovation of Agricultural Science and Technology in Jiangsu, http://czt.jiangsu.gov.cn/, Funding acquisition, Methodology, Resources, Editing

Shengping LIU, CAAS-ASTIP-2016-AII，Innovation Project of the Chinese Academy of Agricultural Sciences，http://www.mof.gov.cn/index.htm，Funding acquisition, Supervision, review & editing”

“This research was supported by the National Natural Science Foundation of China (32172110,31701355), Fundamental Scientific Research Business Expenses of Central Public Welfare Research Institutes(JBYW-AII-2020–29), Jiangsu Key Research and Development Program(BE2020319-14), Special Fund for Independent Innovation of Agricultural Science and Technology in Jiangsu(CX(21)3065，CX(21)3063), Innovation Project of the Chinese Academy of Agricultural Sciences(CAAS-ASTIP-2016-AII).”

“Xiaochun ZHONG, JBYW-AII-2020–29，Fundamental Scientific Research Business Expenses of Central Public Welfare Research Institutes，http://www.mof.gov.cn/index.htm， Conceptualization, Funding acquisition, Review

TaoLIU, 32172110,31701355, National Natural Science Foundation of China, https://www.nsfc.gov.cn/english/site_1/index.html,Funding acquisition, Methodology, Resources, Editing

TaoLIU, BE2020319-14, Jiangsu Key Research and Development Program, http://czt.jiangsu.gov.cn/, Funding acquisition, Methodology, Resources, Editing

TaoLIU, CX(21)3065，CX(21)3063, Special Fund for Independent Innovation of Agricultural Science and Technology in Jiangsu, http://czt.jiangsu.gov.cn/, Funding acquisition, Methodology, Resources, Editing

Shengping LIU, CAAS-ASTIP-2016-AII，Innovation Project of the Chinese Academy of Agricultural Sciences，http://www.mof.gov.cn/index.htm，Funding acquisition, Supervision, review & editing”

“NO authors have competing interests”

**Additional Editor Comments:**

The manuscript is dealing with very important subject that is extracting data using images. II have myself also read the manuscript and found this very useful for the scholars/students working in the area. The manuscript needs improvement by keeping in view comments of the reviewer. Some of my own comments are:

1. Improve presentation of data in tables and figures

2. The different steps involved may be presented in the form of flow chart

3. Test files may be provided in the form of supplementary material

4. Language and grammatical issues could be rectified.

**Reviewers' comments:**

Reviewer's Responses to Questions

**Comments to the Author**

1. Is the manuscript technically sound, and do the data support the conclusions?

Reviewer #1: Yes

2. Has the statistical analysis been performed appropriately and rigorously? 

Reviewer #1: Yes

3. Have the authors made all data underlying the findings in their manuscript fully available?

Reviewer #1: Yes

4. Is the manuscript presented in an intelligible fashion and written in standard English?

Reviewer #1: Yes

5. Review Comments to the Author

Reviewer #1: This study proposes a grain counting algorithm with a wide range of applications, which supports the calculation of the number of grains on white paper by taking pictures of mobile phones. The method proposed in this paper has low cost, simple operation and strong applicability. The problems are as follows:

1.Special perspective analysis section, How to explain the missed detection and error of feature points in the sample diagram, it is recommended to expand the description.

2.The code of the volatility is inconsistent with the calculation formula in Evaluation index.

3.In efficiency comparison section,there is no source for the counting machine with mechanical rotation takes 24 s.

4.The reference format of the two documents such as 31-32 is inconsistent with others.

5.It is recommended to remove the content of the lightbox and focus on analyzing the white paper background.

6.Capitalize the first letter in Figure S2.

7.In the evaluation of identification efficiency, "(3) The analysis time did not maintain a linear positive correlation with the number of grains in the picture. There were various factors that affected the running time, such as the tightness of the grains, size of the picture, and shape of the grains., etc." The conclusion of this sentence should be supported by corresponding experiments or cited.

8.Pay attention to the expression of some phrases. Thousand grain, over segmentation, open source, short axis, etc. should have hyphens, "mega pixels" should be changed to "megapixels"; "light boxes" should be changed to "lightboxes".

9.Pay attention to plurals and grammar.such as“0.7 second”changed to“0.7 seconds”, "provide power support" changed to "provide powerful support" , "robust to variety changes" changed to "robust to various changes".

10.Pay attention to the expressions of some statements, such as "There were various factors that affected the running time" changed to "various factors affected……”, "as Lines …… show in Fig 4" changed to "as Lines …… shown in Fig 4".

11.The sentence is not clear: To improve the calculation efficiency and shorten the response time, the original image was specially compressed, but excessive compression affected the accuracy of the algorithm.

6. PLOS authors have the option to publish the peer review history of their article (what does this mean?). If published, this will include your full peer review and any attached files.

Reviewer #1: No

---

## [Author Response · Author response to Decision Letter 0]

13 Jul 2022

Response to Reviewers

ACADEMIC EDITOR:

The manuscript is dealing with very important subject that is extracting data using images. II have myself also read the manuscript and found this very useful for the scholars/students working in the area. The manuscript needs improvement by keeping in view comments of the reviewer. Some of my own comments are:

1.Improve presentation of data in tables and figures

Re: have modified Tables 1 - 5, Fig 2, Fig 7, Fig9, Fig11, Fig 12. The serial number has changed because Fig 10 has been removed.

2. The different steps involved may be presented in the form of flow chart

Re: Adjusted, demonstrated by flow chart.

3. Test files may be provided in the form of supplementary material

Re: The test images have been packaged and used as supplementary material, it contains test images of 6 different grains (144) and test results (excel). It is relatively large, with 460M of storage space.

4. Language and grammatical issues could be rectified.

Re: We have let an expert who is good at English to revise it. Please see the latest manuscript.

Reviewer #1: 

This study proposes a grain counting algorithm with a wide range of applications, which supports the calculation of the number of grains on white paper by taking pictures of mobile phones. The method proposed in this paper has low cost, simple operation and strong applicability. The problems are as follows:

1.Special perspective analysis section, How to explain the missed detection and error of feature points in the sample diagram, it is recommended to expand the description.

Re: Modified as suggested. Added example descriptions and also labeled the images. The description is as follows.

As shown in Fig 7, where there was no Cpeak point above due to overlap, and only the Cpeak point below was detected, and calculation result of the corner point algorithm was 1.5, which was actually 2 grains.

2.The code of the volatility is inconsistent with the calculation formula in Evaluation index.

Re: Adjusted, the code should be S2, not S2.

3.In efficiency comparison section,there is no source for the counting machine with mechanical rotation takes 24 s.

Re: Adjusted, added citation to indicate the source of the data

4.The reference format of the two documents such as 31-32 is inconsistent with others.

Re: Adjusted, removed the citation article type

5.It is recommended to remove the content of the lightbox and focus on analyzing the white paper background.

Re: Adjusted, removed the contrast with the lightboxes as the background.

6.Capitalize the first letter in Figure S2. 

Re: Adjusted, please see the updated picture.

7.In the evaluation of identification efficiency, "(3) The analysis time did not maintain a linear positive correlation with the number of grains in the picture. There were various factors that affected the running time, such as the tightness of the grains, size of the picture, and shape of the grains., etc." The conclusion of this sentence should be supported by corresponding experiments or cited.

Re: Adjusted, added citation support statement.

8.Pay attention to the expression of some phrases. Thousand grain, over segmentation, open source, short axis, etc. should have hyphens, "mega pixels" should be changed to "megapixels"; "light boxes" should be changed to "lightboxes".

Re: Adjusted as recommended.

9.Pay attention to plurals and grammar.such as“0.7 second”changed to“0.7 seconds”, "provide power support" changed to "provide powerful support" , "robust to variety changes" changed to "robust to various changes".

Re: Adjusted as recommended.

10.Pay attention to the expressions of some statements, such as "There were various factors that affected the running time" changed to "various factors affected……”, "as Lines …… show in Fig 4" changed to "as Lines …… shown in Fig 4".

Re: Adjusted as recommended.

11.The sentence is not clear: To improve the calculation efficiency and shorten the response time, the original image was specially compressed, but excessive compression affected the accuracy of the algorithm.

Re: Modified as suggested, as follows:

The resolution of the original photo was so large that uncompressed operations would greatly increase the algorithm processing time, but excessive compression would reduce the algorithm accuracy. To balance the accuracy and response time, the following conventional resolutions were tested: 1080, 1440, 1920, 2500, and 3000.

---

## [Editor Report · Decision Letter 1]

16 Aug 2022

Research on a rapid identification method for counting universal grain crops：a fast, accurate, low-cost, easy-to-operate, and universal grain counting tool

PONE-D-21-36225R1

Dear Dr.Xiaochun zhong,

We’re pleased to inform you that your manuscript has been judged scientifically suitable for publication and will be formally accepted for publication once it meets all outstanding technical requirements.

Kind regards,

Reyazul Rouf Mir, PhD

Academic Editor

PLOS ONE

Additional Editor Comments (optional):

I am very happy to notice that authors has addressed the comments of the reviewer including my own comments in the revised version f the manuscript. Therefore, I have no hesitation in recommending acceptance of this manuscript.
---

## [Editor Report · Acceptance letter]

22 Aug 2022

PONE-D-21-36225R1 

Research on a rapid identification method for counting universal grain crops 

Dear Dr. Zhong:

I'm pleased to inform you that your manuscript has been deemed suitable for publication in PLOS ONE. Congratulations! Your manuscript is now with our production department. 

Kind regards, 

on behalf of

Dr. Reyazul Rouf Mir 

Academic Editor

PLOS ONE